# Structure and function of the SIT1 proline transporter in complex with the COVID-19 receptor ACE2

Huanyu Z. Li[1], Ashley C. W. Pike[1], Irina Lotsaris[2], Gamma Chi[1], Jesper S. Hansen[1], Sarah C. Lee[3], Karin E. J. Rödström[1], Simon R. Bushell[1], David Speedman[1], Adam Evans[1], Dong Wang[1], Didi He[1], Leela Shrestha[1], Chady Nasrallah[1], Nicola A. Burgess-Brown[1], Robert J. Vandenberg[2] ✉, Timothy R. Dafforn[3] ✉, Elisabeth P. Carpenter[1] ✉ & David B. Sauer[1] ✉

Proline is widely known as the only proteogenic amino acid with a secondary amine. In addition to its crucial role in protein structure, the secondary amino acid modulates neurotransmission and regulates the kinetics of signaling proteins. To understand the structural basis of proline import, we solved the structure of the proline transporter SIT1 in complex with the COVID-19 viral receptor ACE2 by cryo-electron microscopy. The structure of pipecolate-bound SIT1 reveals the specific sequence requirements for proline transport in the SLC6 family and how this protein excludes amino acids with extended side chains. By comparing apo and substrate-bound SIT1 states, we also identify the structural changes that link substrate release and opening of the cytoplasmic gate and provide an explanation for how a missense mutation in the transporter causes iminoglycinuria.

Proline is the only amino acid incorporated into proteins that lacks a primary amine group. With its pyrrolidine ring, the amino acid's restricted Ramachandran angles and hydrogen bonding capability have pronounced effects on polypeptide secondary structure[1]. Consequently, the residue is usually found at the ends of alpha helices and at bends in helices where it disrupts the hydrogen bonding pattern[2], while proline and its derivative hydroxyproline are overrepresented in Polyproline-II helices and the collagen triple helix[3,4]. Within a protein, proline's unique cis/trans isomer energetics and isomerization kinetics are central to kinetic switches in signaling proteins[5,6], and protein folding[7]. Physiologically, the amino acid also acts as a weak agonist of glycine and ionotropic glutamate receptors[8,9], and hyperprolinemia is associated with autism spectrum disorder, intellectual disability, and psychosis spectrum disorders[10].

Several plasma membrane transporters import proline into the cell, including the Sodium/imino-acid transporter 1 (SIT1) encoded by the SLC6A20 gene. SIT1 was first identified as a proline transporter in

the kidney[11–13]. Consequently, polymorphisms in SLC6A20 lead to iminoglycinuria[14], and are correlated with altered concentrations of secondary and tertiary amine metabolites in plasma and urine[15,16]. Neurologically, SIT1 regulates proline concentrations to modulate the activity of glycine and NMDA-type glutamate receptors in mice[9], and the absence of neurons in the colon, causing Hirschsprung's disease, is associated with SLC6A20 polymorphisms[17–19]. In the eye, SIT1 expression is a signature of the retinal pigment epithelium and drives the proline-preferring metabolism of these cells[20–22]. Accordingly, gene variants are correlated with both retinal and macular thickness, and degenerative macular disease[21,23,24]. Finally, SIT1 traffics to the plasma membrane in a complex with ACE2[25], the SARS-CoV2 receptor. SIT1 overexpression can prevent ACE2 trafficking to the plasma membrane[26] and polymorphisms in the transporter gene are associated with clinical outcomes of SARS-CoV2 infection[27–29].

SIT1 belongs to the SLC6 gene family of amino acid and amine transporters[30], and the larger Neurotransmitter Sodium Symporter

[1]Centre for Medicines Discovery, Nuffield Department of Medicine, University of Oxford, Oxford, UK. [2]Molecular Biomedicine Theme, School of Medical Sciences, University of Sydney, Sydney, NSW, Australia. [3]School of Biosciences, University of Birmingham, Birmingham, UK. ✉e-mail: robert.vandenberg@sydney.edu.au; t.r.dafforn@bham.ac.uk; lizcarpen1@gmail.com; david.sauer@cmd.ox.ac.uk

(NSS) superfamily. The structure, selectivity, and transport for SLC6 and NSS transporters has been revolutionized by structural studies of the prokaryotic amino acid transporters LeuT and MhsT[31-36], structurally homologous bacterial transporters Mhp1 and vSGLT[37,38], and several eukaryotic SLC6 transporters[39-43]. Central to NSS-mediated substrate transport is the LeuT protein fold, a compact domain composed of 10 transmembrane helices[31]. Within this structure, substrates and co-transported ions bind at sites created by breaks in TM1 and TM6, and differences in sequence within and near this region determine the proteins' substrate selectivity[35,36]. Starting in an outward-open apo state, substrate and ion binding induce closure of the extracellular gate, through the movement of TM1b and TM6a and residues lining those helices which block access to the binding site[31,34]. From this occluded state, the cytoplasmic gate subsequently opens through tilting of TM1a into the plane of the bilayer[34,44]. Coupled to the tilting of TM1a are movements on the cytoplasmic face of the protein, particularly in gating helix TM5 with its highly conserved $GX_NP$ motif[32].

SIT1 and PROT (SLC6A7) are unique among the SLC6 family in preferring amino acids with a secondary amine[11-13,45], though other SLC6 transporters can transport both primary and secondary amino acids[30]. While the mechanism of substrate selectivity within the SLC6 family is of great interest, the proline transporters are relatively understudied. Furthermore, low sequence similarity limits useful comparison of SIT1 to the prokaryotic proline transporter PutP despite similar substrate selectivity profiles[46] (Supplementary Fig. 1a). Therefore, while homology models based on the prokaryotic LeuT structure have been used to probe SIT1's ion binding[47], the mechanism for its selective transport of secondary amino acids remains unclear.

In this study, we probe SIT1's selectivity and transport mechanism with a combination of thermostabilization-based SIT1 binding assays, electrophysiological analysis of ion-coupled transport and cryo-electron microscopy (cryo-EM) based structural studies of the ACE2-SIT1 complex. From these results, we propose a structural model for SIT1's preference for secondary amino acids and the conformational changes underlying amino acid release.

## Results

### Structure of ACE2-SIT1 complex

After overexpressing and purifying SIT1 (Supplementary Fig. 1b, c), we first validated substrate binding of SIT1 in detergent (Fig. 1a, Supplementary Fig. 1d). In agreement with the transporter's in vivo selectivity[13], proline and pipecolate increased the protein melting temperature ($T_M$) by 3 °C and 6 °C, respectively, while glycine and sarcosine had no apparent effect. Aiming to examine the structural interactions of SIT1 and substrate, the small size of the transporter-amino acid complex presented a challenge for single-particle cryo-EM. To increase the particle mass, we expressed and purified SIT1 in complex with its trafficking chaperone ACE2 (Supplementary Fig. 1e, f)[25], a strategy also used in determining the structure of apo-SIT1 and the neutral amino acid transporter B⁰AT1[26,39-41].

Single-particle cryo-EM analysis ultimately yielded a nominal 3.24 Å map of the ACE2-SIT1 complex, determined in the presence of pipecolate (Fig. 1b, Supplementary Fig. 2a, Table 1). This map was sufficiently detailed to model residues 10–582 of SIT1 and 21–768 of ACE2. SIT1 adopts the classic LeuT-fold expected for this family of amino acid transporters (Supplementary Fig. 3a, b)[26,47], while ACE2 is composed of peptidase (PD) domain and collectrin-like domain with transmembrane and neck regions[39]. The SIT1 transporter is N-glycosylated at Asn131 and Asn357 (Fig. 1c, Supplementary Fig. 3f). ACE2 is more heavily decorated[48], with visible N-linked glycans at Asn90, Asn103, Asn322, Asn432, Asn546, Asn690, and an O-glycosylation at Thr730 (Fig. 1c, Supplementary Fig. 3f). Most of these glycan chains do not significantly interact with the protein, and consequently only a single sugar is resolvable. However, a branched N-linked glycan at ACE2's Asn690 extensively hydrogen bonds with the peptidase domain (Fig. 1f, g).

As with the homologous ACE2-B⁰AT1 complex, the structure is a dimer of heterodimers where homodimerization of ACE2 is mediated primarily by its neck domain while the ACE2 and transporter subunits interact via three distinct sets of contacts (Fig. 1c, Supplementary Fig. 3c-e). On the extracellular side of the membrane, the C-terminal

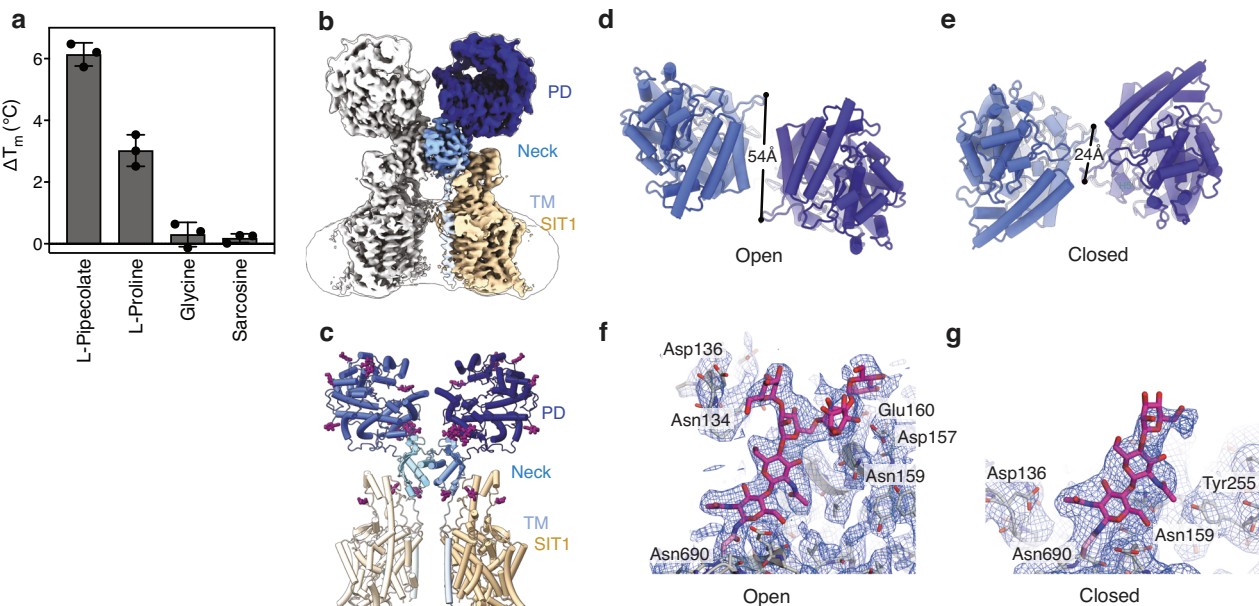

**Fig. 1 | Structure of the ACE2-SIT1 complex determined in the presence of pipecolate by cryo-EM. a** Change in melting temperature of SIT1 by amino acids. Data are presented as mean ± SEM, with individual data shown as circles ($n = 3$). **b** Cryo-EM map of the ACE2-SIT1 complex determined in the presence of pipecolate. The ACE2 peptidase, neck, and TM domains are colored blue, cyan, and light blue, respectively. SIT1 is colored in wheat. Overlayed semi-transparent is the same map, low-pass filtered, showing the entire particle including the detergent micelle. **c** Protein structure of the ACE2-SIT1 complex, viewed from the plane of the membrane. Glycans are shown as purple sticks. ACE2's peptidase domain in **d** open and **e** closed conformations. The glycan chain at Asn690, viewed from the membrane surface, interacting with the **f** open and **g** closed conformations of the peptidase domain with Coulombic potential maps shown as mesh.

**Table 1 | CryoEM data collection, processing, refinement and model validation**

| | Pipecolate-bound | | | Apo | | |
|---|---|---|---|---|---|---|
| | PD Open | PD Closed | SIT1 focused | PD Open | PD Closed | SIT1 focused |
| EMDB ID | EMD-17381 | EMD-17382 | EMD-17380 | EMD-17378 | EMD-17379 | EMD-17377 |
| PDB ID | 8P30 | 8P31 | 8P2Z | 8P2X | 8P2Y | 8P2W |
| Data collection: | | | | | | |
| Microscope | eBIC Krios II | | | OPIC Krios | | |
| Camera | Gatan K3 | | | Gatan K2 | | |
| Nominal Mag | 105,000 | | | 165,000 | | |
| Pixel size (Å/px) | 0.831 | | | 0.82 | | |
| Total Dose (e-/Å$^2$) | 50 | | | 50.09 | | |
| Defocus range (μm) | −1.2 to −2.4 | | | −1.0 to −2.2 | | |
| Micrographs collected | 13,194 | | | 8704 | | |
| Initial Particles after 2D | 808,912 | | | 509,681 | | |
| Final Particles | 206,023 | 154,699 | 344,606 | 136,985 | 96,564 | 322,202 |
| Symmetry imposed | C2 | C2 | C1 | C2 | C2 | C1 |
| Map resolution (Å) (GSFSC = 0.143) | 3.29 | 3.24 | 3.49 | 3.59 | 3.46 | 3.76 |
| Refinement: | | | | | | |
| Resolution (Å) | 3.3 | 3.3 | 3.5 | 3.6 | 3.5 | 3.8 |
| Sharpening B-factor (Å$^2$) | −75 | −75 | −112.5 | −147.4 | −124.1 | −152.2 |
| Map CC (phenix CC mask) | 0.8082 | 0.8319 | 0.8034 | 0.7946 | 0.7796 | 0.7671 |
| Map-to-model resolution (Å) (FSC = 0.5) | 3.51 | 3.42 | 3.64 | 3.77 | 3.74 | 4.01 |
| Model composition/validation: | | | | | | |
| Non-hydrogen atoms | 21,074 | 20,752 | 4641 | 20,828 | 20,902 | 4526 |
| Protein residues | 2680 | 2670 | 606 | 2674 | 2668 | 603 |
| Ligands | 10 | 10 | 10 | | | |
| Rms bonds (Å) | 0.003 | 0.003 | 0.003 | 0.002 | 0.002 | 0.003 |
| Rms angles (°) | 0.510 | 0.552 | 0.545 | 0.409 | 0.397 | 0.443 |
| Molprobity Score | 1.46 | 1.55 | 1.54 | 1.39 | 1.38 | 1.68 |
| Clash Score | 6.06 | 6.85 | 7.41 | 5.24 | 5.39 | 5.43 |
| Rotamer outliers (%) | 0.23 | 0.58 | 0.21 | 0.48 | 0.47 | 1.79 |
| Ramachandran (favored) % | 97.30 | 96.96 | 97.32 | 95.85 | 97.57 | 96.82 |
| Ramachandran (allowed) % | 2.70 | 3.04 | 2.68 | 4.15 | 2.43 | 3.18 |

portion of TM7 and loop prior to EH5 from SIT1 hydrogen bond with the collectrin-like domain of ACE2. SIT1's ECL2 also hydrogen bonds with the extended region between the neck and TM domain of ACE2. Within the membrane, the transmembrane helix of ACE2 makes extensive van der Waals contacts with TM3 and TM4 of SIT1. Notably, these structural features of SIT1 correspond to sequence motifs conserved in proteins that form complexes with ACE2. In particular, the TM7 extension is specific to the ACE2-interacting SIT1, B⁰AT1, and B⁰AT3[49,50]. Similarly, the ECL2 sequence is highly conserved within these three SLC6 proteins. Furthermore, SIT1's Leu183 and Leu186 on TM4 directly contact the transmembrane helix of the chaperone. Equivalent residues in the dopamine and GABA transporters, which do not form stable complexes with ACE2[42,51], are phenylalanine or tryptophan, respectively. Therefore, we hypothesize larger residues at these positions are a steric barrier to binding ACE2 or collectrin, and thereby partially explain the binding of the chaperones by particular members of the SLC6 family.

While processing and classifying particles from the ACE2-SIT1 sample, obvious structural and compositional heterogeneity was apparent within the dataset (Supplementary Fig. 2a). Unlike previous ACE2 complex structures, this ACE2-SIT1 dataset yielded maps with 2:2 and 2:1 stoichiometry of ACE2 to transporter, with 3-fold more particles in the larger complex. This varying stoichiometry is not due to insufficient SIT1 as un-bound transporter was apparent during purification (Supplementary Fig. 1f). We hypothesize this smaller complex

may be a consequence of low affinity between the SIT1 and ACE2 leading to varying stoichiometry not resolvable by preparative size exclusion chromatography. Alternatively, this structural heterogeneity may be due to the denaturation of a single transporter subunit at the air-water interface[52].

Significant structural heterogeneity in the ACE2's peptidase domain was present within the ACE2-SIT1 complex (Supplementary Fig. 2a). Further classifying the 2:2 ACE2-SIT1 complex data, there are two distinct conformations based on the relative domain orientations within the ACE2 dimer. This movement between open and closed conformations occurs as a 30° rigid-body rotation around a hinge at residues 612-617 in the loop between peptidase and collectrin-like domains (Supplementary Fig. 3g, h). As with the ACE2-B⁰AT1 structure[39], a second dimer interface is formed by Gln139 and Gln175 within the peptidase domain in ACE2's closed conformation (Fig. 1d, e). Further, the branched N-glycan chain at ACE2's Asn690 is rigid and well-ordered in the open conformation, hydrogen bonding with asparagine, aspartate, and glutamate side chains on the peptidase domain's H6 and beta-hairpin between H4 and H5 (Fig. 1f). These interactions break during the PD's conformational change, and fewer protein-glycan contacts are present in the closed conformation (Fig. 1g). Accordingly, the glycan density is weaker in this conformation as the chain becomes more mobile. This suggests that the Asn690 glycan chain acts as a latch on the peptidase domain, stabilizing the open conformation and thereby regulating its change between conformations.

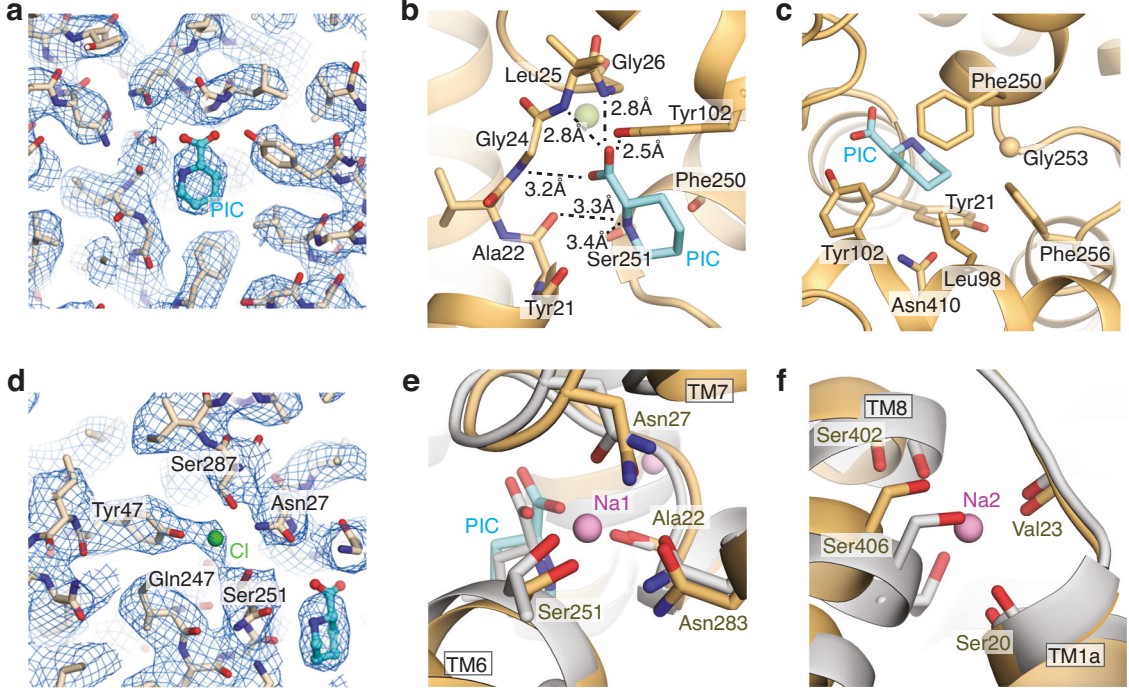

**Fig. 2 | Amino acid, chloride, and sodium binding sites of SIT1. a** Amino acid binding site of SIT1. Coulombic potential map shown as mesh, pipecolate shown as cyan sticks. SIT1's coordination of the pipecolate's (**b**) amine and carboxylate groups and (**c**) piperidine ring. **d** Chloride binding site of SIT1. Coulombic potential map is shown as mesh. Overlay of the (**e**) Na1 and (**f**) Na2 binding sites for SIT1 and LeuT. SIT1 and LeuT are shown in wheat and gray, respectively. Sodium ions from the LeuT structure (2A65) are shown as purple spheres.

## Substrates binding within SIT1

While the 2:2 ACE2-SIT1 complex with the peptidase domain in the closed conformation yielded the highest-resolution map for the entire complex, the map is best in ACE2's neck region and less detailed in the transporter (Supplementary Fig. 2a). We therefore combined particles from closed and open conformations, symmetry expanded over the transmembrane region of ACE2 and SIT1, and performed a further round of 3D classification and local refinement. This strategy produced a 3.49 Å map of the transmembrane domain. Map interpretability improved for the peptide backbone and side chains (Supplementary Fig. 2b–d), enabling more accurate placement for residues 10–582 of SIT1 and 740–768 of ACE2's TMD.

SIT1 is in an occluded conformation with the active site inaccessible to both the cytoplasm and extracellular space. The extracellular gate is closed with Tyr102 and Phe250 blocking access to the substrate binding site. The extracellular path to the substrate site is further obstructed by Tyr33 and Asn461, which are hydrogen bonded, and potential water-mediated hydrogen bonds between Arg30, Tyr38, Asn243, and Asp462. On the opposite side of the membrane, the cytoplasmic gate formed by TM1a is stabilized primarily through van der Waals interactions with TM6b and TM7. This gate is further held closed by two hydrogen bond networks linking TM1a's Ser11 with Asn270 and His275 on TM6, and Tyr21 with Gly253 and Ser258 on TM7 and Asn410 on TM8.

Within the SIT1 binding site are two non-protein densities, identified as pipecolate and chloride based on size, local chemistry, and similarity to structures of other LeuT-fold transporters (Fig. 2a, d). The Coulombic potential map for pipecolate envelops the piperidine ring, with only the partial coverage of the carboxylate moiety. This is unsurprising as electrons scatter more weakly from negatively charged atoms and carboxylates are more radiation sensitive[53–55]. Nevertheless, this density and the local chemistry allow us to unambiguously place pipecolate, observing SIT1 engages the distinct chemical moieties of the ligand through three regions of the binding site. The pipecolate's

amino group is surrounded by the carbonyls of Tyr21 and Ala22 of TM1 and Phe250 and Ser251 of TM6 (Fig. 2b), with Ala22 and Ser251 best placed for hydrogen bonds. The substrate's carboxyl moiety interacts with the amide nitrogens of Gly24, Leu25, and Gly26, and the side chain hydroxyl from Tyr102 of TM3 (Fig. 2b). Finally, the substrate's piperidine ring is coordinated by van der Waals contact from Tyr21 of TM1, Leu98 of TM3, Phe250, Gly253, and Phe256 of TM6, and Asn410 of TM8 (Fig. 2c).

SIT1's binding of pipecolate is very similar to the amino acid coordination by the S1 binding site of LeuT and MhsT[31,32] (Fig. 3a, Supplementary Fig. 4c, f). In contrast, the pose of leucine in the related B⁰AT1's structure is significantly different from that observed here, or in LeuT and MhsT (Supplementary Fig. 4a, d). However, the leucine-bound ACE2-B⁰AT1-focused cryo-EM map (EMD-30043) contains ambiguous densities for the substrate. Refitting the leucine and binding site residues in B⁰AT1 to positions more consistent with homologous transporters (Supplementary Fig. 4b, e) yielded an improved substrate FSC-Q[56]. Therefore, we used this re-refined model of ACE2-B⁰AT1 bound to leucine for all subsequent comparisons.

As a sodium and chloride coupled co-transporter, SIT1 has obvious binding sites for two sodium and one chloride ion. The density for the Cl⁻ is clear in the experimental Coulombic potential map (Fig. 2d). Within SIT1, the anion is coordinated by the side chains of Asn27 of TM1, Tyr47 of TM2, Gln247 and Ser251 of TM6, and Ser287 of TM7 in a mode very similar to hSERT and the engineered, chloride-dependent LeuT[43,57]. Notably, there is no apparent density within SIT1 for sodium ions at the expected Na1 and Na2 sites, despite an experimental concentration 7-fold greater than the cation's $K_M$[47]. However, sodium density is often weak or absent in cryo-EM maps at similar or better resolution[58–60]. Nevertheless, the coordinating moieties from SIT1 and substrate are oriented similarly to the sodium-bound state of LeuT (Fig. 2e, f). Furthermore, the valence at the Na1 ($v_{Na} = 2.5$) and Na2 ($v_{Na} = 0.41$) sites indicate reasonable coordination for sodium ions[61]. Therefore, we propose this structure captures the sodium, chloride,

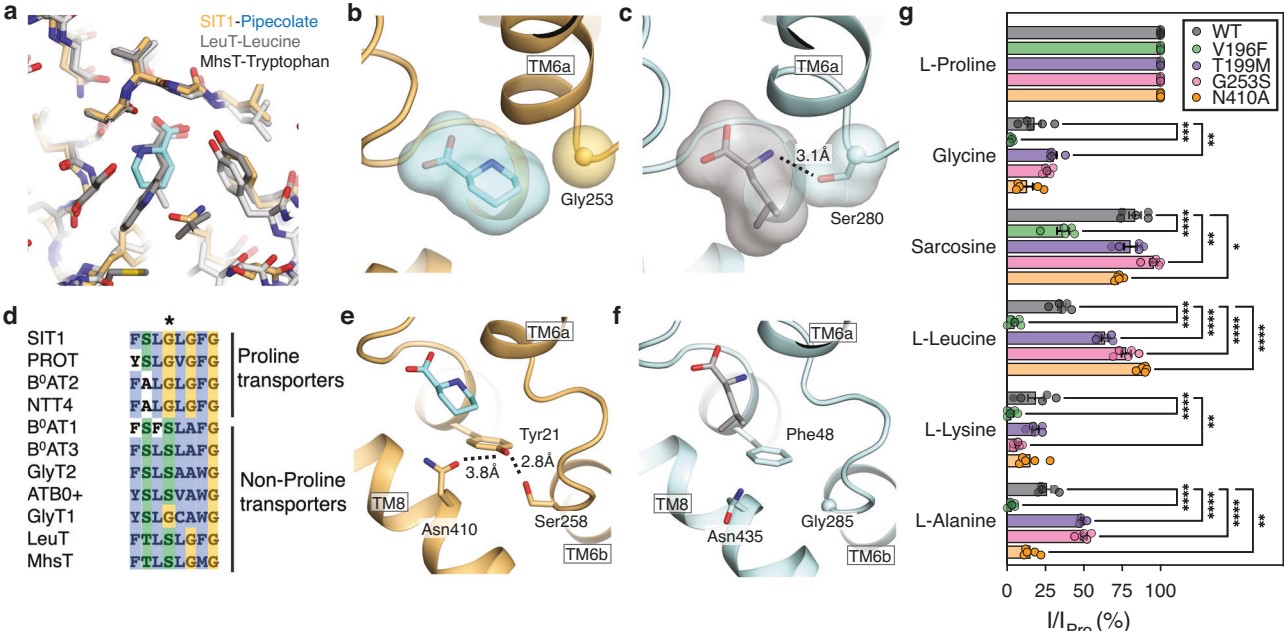

**Fig. 3 | Binding site differences between proline selective and non-selective SLC6 amino acid transporters. a** Overlay of amino acid bound structures of SIT1, LeuT (2A65), and MhsT (4US4). Binding site for **b** SIT1 and **c** the refit B⁰AT1, colored in wheat and light blue, respectively. Substrate pipecolate and leucine are shown in cyan and gray, respectively. **d** Sequence alignment of SLC6 amino acid transporters. The position of SIT1's Gly253 is indicated. Hydrogen bond network of the pipecolate coordinating residues Tyr21 and Asn410 (**e**) in SIT1 and (**f**) equivalent

region in B⁰AT1. **g** Relative amino acid transport of SIT1 and its mutants. *Xenopus laevis* oocytes expressing mutant or WT SIT1 were perfused with 1 mM L-proline, glycine, sarcosine, L-leucine, L-lysine or L-alanine. Transport currents are normalized to the current generated by 1 mM L-proline. Data are presented as mean ± SEM, with individual data shown as circles ($n = 5$). Significance (2-way ANOVA) is denoted by *, where *$p < 0.05$, **$p < 0.01$, ***$p < 0.001$, and ****$p < 0.0001$.

and substrate-bound inward-facing occluded state of SIT1's reaction cycle.

## Proline binding in the SLC6 family

To understand the capability and preference for SLC6 transporters to import proline, we compared the binding site interactions of SIT1 to the structures and sequences of related transporters. As previously noted, Gly253 packs immediately against the piperidine ring of pipecolate (Fig. 3b), and other SLC6 transporters of proline all possess a glycine at the equivalent position (Fig. 3d). In contrast, nearly all of the remaining amino acid transporting SLC6s have a serine at this position, which make hydrogen bonds with the substrates' primary amine in B⁰AT1, LeuT, MhsT (Fig. 3c, Supplementary Fig. 4g)[31,32]. Such a serine likely stabilizes the binding of a primary amino acid, and correspondingly introducing the mutation G253S to SIT1 increases its relative transport of L-leucine and L-alanine (Fig. 3g). Therefore, we propose that SLC6 proteins that preferentially transport proline require a glycine at this position to reduce their affinity for primary amino acids and thereby selectively transport proline.

While SIT1's Gly253 explains a necessary component of secondary amine transport, we also noted a second structural feature that explains SIT1's exclusion of amino acids with extended side chains[13]. Within the substrate binding site, pipecolate is contacted by the side chains of Tyr21 and Asn410, which are in a hydrogen bond network with Ser258 (Fig. 3e, Supplementary Fig. 4g). This network restrains the position of Asn410 such that it would clash with amino acid substrates possessing extended side chains. This immediately suggests a mechanism for SIT1's selectivity, where Asn410 is a steric block to exclude substrate amino acids with extended sidechains. Supporting this model, we found the SIT1 mutation N410A significantly increases the transport of L-leucine (Fig. 3g), likely by alleviating this steric block. This model for amino acid selectivity is also consistent with sequence variation across the SLC6 family, where the neutral amino acid

transporting B⁰AT1 lacks this hydrogen bond, with phenylalanine at the position equivalent to SIT1's Tyr21. Therefore, B⁰AT1's equivalent asparagine, Asn435, is free to move and thereby accommodate the substrate's extended side chain, as seen in protein's leucine-bound structure (Fig. 3f, Supplementary Fig. 4g). Furthermore, PROT also appears to use a similar steric block to exclude extended side chain substrates. Rather than the rigidly oriented asparagine of SIT1, in PROT the bulk of a phenylalanine at the equivalent position to Asn410 prevents access to the side chain pocket (Supplementary Fig. 4h).

## Opening of SIT1's cytoplasmic gate in the absence of substrate

Having captured the inward-facing occluded state of SIT1's reaction cycle, we next set out to characterize the protein's conformational changes upon substrate release. We therefore determined the ACE2-SIT1 structure, with an open-conformation peptidase domain, in the presence of glycine to a resolution of 3.46 Å overall and 3.76 Å for the transporter alone (Supplementary Fig. 5, Supplementary Fig. 6a). The maps were sufficiently detailed to model and refine residues 11–582 of SIT1 and 20–768 of ACE2 (Supplementary Fig. 6b–d). This SIT1-structure agrees well with the recently published ACE2-SIT1 structures in complex with receptor binding domains from SARS-CoV2 and determined in amino acid-free buffers (RMSD = 0.765–0.940).

Without a secondary amino acid substrate, SIT1 has undergone structural rearrangements that open the transporter's cytoplasmic gate (Fig. 4a, b). The greatest movement is a rigid body 17° tilt of TM1a (Fig. 4c). This movement breaks most of TM1a's closed-conformation hydrophobic interactions with TM6b and TM7, and the hydrogen bonds of Ser11 (Fig. 4d). The absence of density for Tyr21 indicates it is mobile and therefore no longer hydrogen bonded to Ser258 on TM6b. Rather, in this inward-open conformation TM1a forms exclusively van der Waals contacts with TM5 and TM7 (Fig. 4e). This agrees with the weaker TM1a interactions in other LeuT-fold transporters upon substrate release which increase the dynamics of this helix[62].

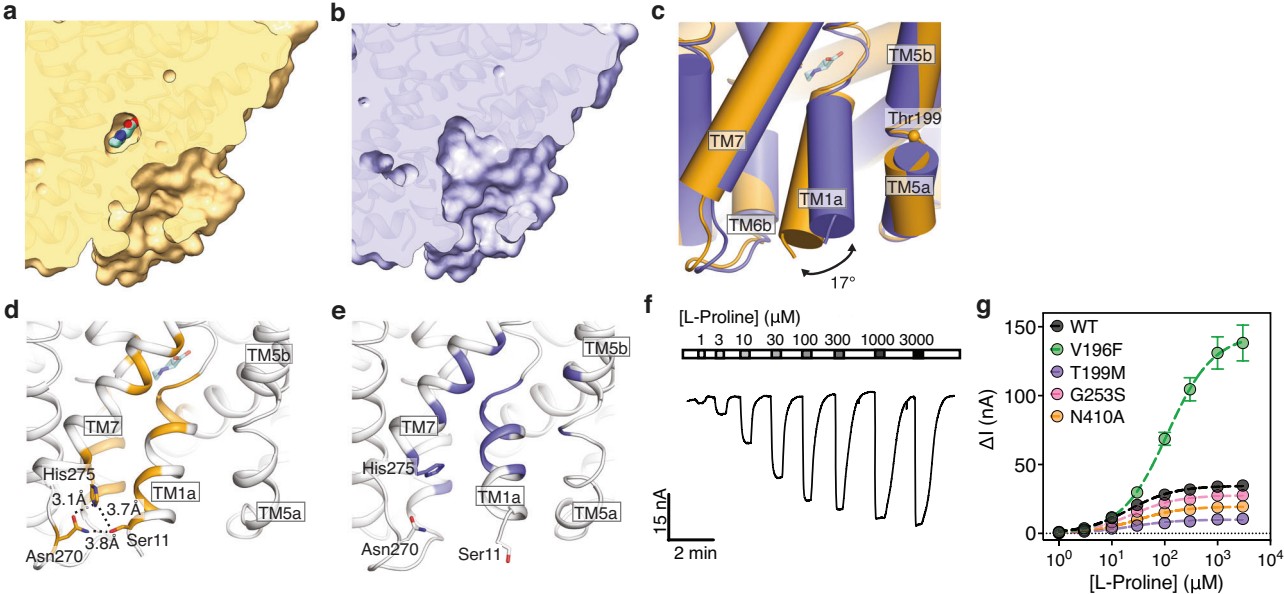

**Fig. 4 | Cryo-EM structure of the ACE2-SIT1 complex determined in the inward-facing apo state.** Cross section of the ACE2-SIT1 transmembrane domain in **a** pipecolate-bound and **b** apo state, determined in the presence of glycine. **c** Overlay of SIT1 in the pipecolate-bound and apo states. **d** The structure of pipecolate-bound SIT1, with residues within 3.9 Å of TM1a highlighted in wheat. **e** The structure of apo SIT1, with residues within 3.9 Å of TM1a shown in purple. **f** An example current trace for an oocyte expressing SIT1 with gray bars representing increasing concentrations of L-proline. **g** L-proline concentration response curves for wild type and mutant SIT1. Data are presented as mean values ± SEM among five independent experiments. The fitted curve was used to determine the $K_m$ and $I_{max}$.

## Mechanism of the iminoglycinuria mutation

The open conformation of SIT1's cytoplasmic gate is stabilized by TM1a's interactions with TM5, which has moved laterally within the membrane (Fig. 4c, e). This mobile portion of TM5 corresponds with the conserved $GX_NP$ motif essential to opening the cytoplasmic gate of LeuT-fold transporters[32]. This immediately suggested a mechanistic explanation for the mutation in TM5, T199M, implicated in iminoglycinuria (Fig. 4c)[14]. We expect the larger methionine side chain in the mutation to interfere with the packing and dynamics of TM5 (Supplementary Fig. 7a, b). This would then alter SIT1's energetics for opening the cytoplasmic gate and thereby reduce the proline transport rate. Supporting this hypothesis, while wild-type SIT1 exhibited robust proline currents in oocytes (Fig. 4f), the T199M mutation produces a 3-fold reduction in SIT1's proline transport $I_{max}$ (Fig. 4g, Table 2). We further probed the role of TM5 in SIT1's gating by mutating Val196, which also interacts with TM1a in the inward-open state. Introducing the larger phenylalanine side chain at this position increases the transporter's $I_{max}$ by 4-fold (Fig. 4g, Table 2), confirming TM1a-TM5 interactions are essential to the transporter's turnover rate. This mechanism for T199M-induced changes to SIT1's transport energetics is also consistent with the partial rescue of SIT1's iminoglycinuria mutant by the secondary mutation M401T[47]. This mutation has been previously proposed to restore activity to the SIT1 T199M mutant by reestablishing a TM8-mediated link between TM5

and the amino acid and sodium sites. However, in our structures there is no change in TM8 upon substrate release. Rather, we propose the M401T mutation, in the background of T199M, allows TM5 to properly pack against TM8 and thereby restores the energetics of opening SIT1's cytoplasmic gate.

## Structural coupling between TM1a and the substrate binding site

Within the SIT1 map determined in glycine, there is no apparent density for amino acid or chloride (Supplementary Fig. 6e, f), though the binding site is similarly resolved to the pipecolate bound structure (Supplementary Fig. 6g, h). Rather, structural changes in this region appear to link the cytoplasmic gate to the sodium, chloride, and substrate sites. Most pronounced, the unwound region of TM1 has repositioned, distorting the hydrogen bond donors and acceptors which previously coordinated the substrate's amine and carboxy groups (Fig. 5a). The side chain of Asn27 has shifted to partially occupy the Na1 site, also preventing its coordination of chloride (Fig. 5b, c). The tilting of TM1a has shifted the carbonyls of Ser20, Ala22, and Val23 by 1.4 to 2.3 Å, disrupting the Na1 and Na2 sites (Fig. 5c, d). These structural distortions reduce the valence at the Na1 ($v_{Na} = 0.26$) and Na2 ($v_{Na} = 0.08$) sites, suggesting that both have lower affinities for sodium. The absence of Na1 may also alter the dynamics of Ser251 which coordinates both that cation and chloride, and an analogous link between substrate, sodium, and chloride binding has been proposed for an engineered LeuT[57].

## Discussion

Here we identify the structural features which determine the SLC6 proteins' capability and preference to transport secondary amino acids, and the structural changes upon substrate release. Specifically, a conserved glycine and asparagine explain SIT1's selectivity for secondary amino acids. Further comparing SIT1's apo and substrate-bound structures, we noted the release of substrate, sodium, and chloride, and the opening of the cytoplasmic gate, are synchronized through modest changes to protein structure in the binding site.

## Table 2 | Proline transport kinetics for SIT1 and mutants

| | Km | *p*-value vs WT[a] | Imax | *p*-value vs WT[a] |
|---|---|---|---|---|
| WT | 21.3 ± 5.1 | | 34.6 ± 1.6 | |
| V196F | 112 ± 19 | <0.001 | 144.4 ± 5.7 | <0.01 |
| T199M | 23.5 ± 7.7 | >0.05 | 10.05 ± 0.64 | <0.01 |
| G253S | 24.7 ± 4.4 | >0.05 | 27.69 ± 0.96 | >0.05 |
| N410A | 29.4 ± 8.3 | >0.05 | 19.5 ± 1.1 | <0.05 |

[a]Statistical significance was determined by Brown-Forsythe and Welch ANOVA tests with Dunnett's T3 multiple comparisons on individual cell data.

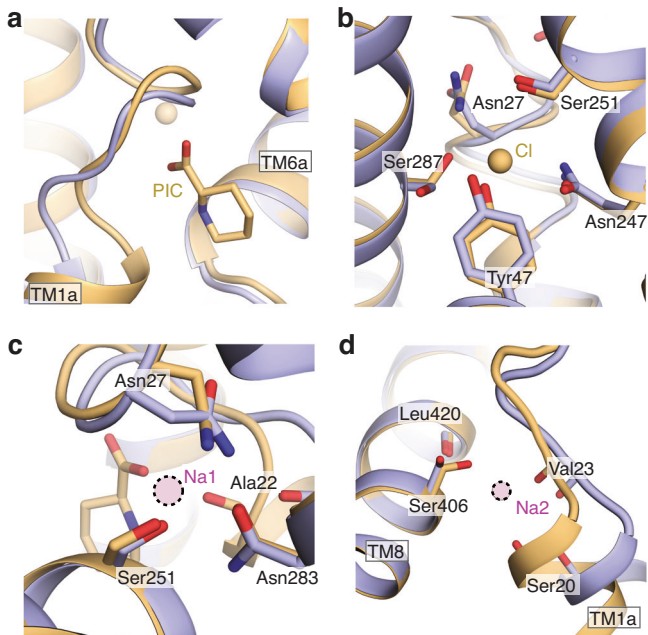

**Fig. 5 | Substrate and ion binding sites of apo SIT1. a** Overlay of SIT1's pipecolate binding site for the bound and apo structures. Pipecolate-bound and apo SIT1 models are colored in wheat and purple, respectively. **b** Overlay of the pipecolate-bound and apo SIT1 chloride binding site. Overlay of pipecolate-bound and apo SIT1 for the (**c**) Na1 and (**d**) Na2 binding sites. Sodium positions are docked from sodium-bound outward-occluded LeuT structure (2A65), and indicated by purple spheres with dotted edges.

While excluding amino acids with extended side chains, SIT1's steric block by Asn410 would allow binding and transport of amino acids with small side chains. This is consistent with the protein's transport of sarcosine (Fig. 3g), and the association of SLC6A20 gene variants with the concentration of dimethylalanine in urine[15]. This steric block also agrees with glycine transport by SIT1, and our results provide additional illumination to the conflicting reports for the transporter's import of that amino acid[9,12,13]. Our biochemical binding results were ambiguous, as both glycine and the bona fide SIT1 substrate sarcosine did not alter the transporter's melting temperature (Fig. 1a, Supplementary Fig. 1d). This demonstrates the transporter's interactions with small amino acids are insufficient to alter apparent stability, and thermostabilization cannot be used to exclude particular amino acids as substrates. However, analysis of the electrophysiology results show that glycine and sarcosine currents were 18% and 83% the proline current. This indicates both molecules are transported but the absence of the N-methyl group causes 4-fold lower $I_{max}$ (Fig. 3g). Finally, our structures revealed a mechanism for the protein's inefficient transport of glycine. SIT1's binding site adopts an inward-open apo state in the presence of that amino acid (Fig. 5b, c), despite being sterically accommodated when docked into the pipecolate-bound SIT1 structure. Notably, when docking sarcosine into the pipecolate-bound structure, its methyl group makes van der Waals contact with Tyr21, while bound glycine would lack this interaction (Figs. 2c and 3e). Therefore, we hypothesize that while glycine can fit within the SIT1 binding site, its primary amine poorly engages with the transporter relative to secondary amine substrates.

Notably, SIT1's structure provides an explanation for the protein's 30-fold selectivity for L-pipecolate and L-proline over their D stereoisomers, while remaining non-selective between N-methyl-L-alanine and N-methyl-D-alanine[11]. Examining the binding pocket, Tyr21, Ser406, and Asn410 would clash with the larger D-amino acids, while N-methyl-D-alanine can be accommodated. This agrees with similar asymmetry in the binding site shape and electrostatics of prokaryotic

LeuT-fold amino acid transporters which drive their amino acid stereoselectivity[63].

The SIT1-pipecolate complex structure also suggests an explanation for the transporter's preference for that amino acid over the smaller proline[11]. Within the substrate binding site, Tyr21's aromatic ring makes a van der Waals contact with the pipecolate (Fig. 3e), and the position of this tyrosine is restrained by the hydrogen bonds with Ser258 and Asn410. Therefore, we hypothesize this positional restraint prevents Tyr21 from making effective van der Waals contacts with smaller proline, explaining SIT1's relative affinity. In contrast, PROT lacks the hydrogen bonding side chains on TM6 and TM8, equivalent to SIT1's Ser258 and Asn410 (Supplementary Fig. 4h). Therefore, PROT's Tyr53 can reposition for van der Waals contacts with smaller cyclic amino acids. Accordingly, PROT has a 2-fold tighter affinity for proline over pipecolate[64].

Finally, our structure and sequence provide hints for SIT1's transport for N-methyl-L-proline[13], while PROT cannot[64]. This permissivity of SIT1 to N-methylated cyclic amino acids is also seen in GWAS analysis of circulating metabolites, where SLC6A20 gene variants are associated with the plasma concentrations of tigonelline and N-methylpipecolate[16]. Notably, within the SIT1-pipecolate structure the substrate's amine nitrogen is engaged by Ala22 (Fig. 2b), and docking N-methylpipecolate into the structure produces a clash with that residue. We speculate the unwound region of TM1 in SIT1 can repack around the methyl groups of tertiary amine substrates, enabled by the small alanine side chain. However, the larger equivalent cysteine of PROT may block this structural accommodation, thereby explaining the proteins' difference in selectivity for N-methylated cyclic amino acids.

Notably, the direct interaction of SIT1's TM1a with substrate also explains the change in selectivity for mutations V196F and T199M on TM5 (Fig. 3g, Table 2), and why $I_{max}$ is altered for binding site mutation N410A (Fig. 4g, Table 2). As this helix forms both the cytoplasmic gate and part of the binding site the open-closed conformational equilibrium of this structural motif will affect, and be affected by, substrate affinity and transport rate.

## Methods

### Ethical statement
*Xenopus laevis* oocytes were supplied by the Victor Chang Cardiac Research Institute and approved by the Garvan Institute/St Vincents Hospital Animal Ethics Committee (Animal Research Authority 23_11).

### Sequence alignment and phylogenomic analysis
SLC6 family protein sequences, and related bacterial homologs, were aligned in Promals3D[65]. Phylogenetic distances were calculated using FastTree 2 with default settings[66], and rooted using NCC1 as the outgroup[67].

### Cloning
The full-length, codon-optimized sequence of human ACE2 was cloned into pHTBV (kindly provided by Prof. Frederick Boyce, Harvard) with N-terminal FLAG tag. The SIT1 sequence was cloned into pHTBV with C-terminal twin-Strep and 10-His tags. Baculoviruses for each construct were generated using standard methods[68]. Baculoviral DNA from transformed DH10Bac was used to transfect Sf9 cells to produce baculovirus particles, which were then amplified with Sf9 cells grown in Sf-900 II medium (Thermo Fisher Scientific) supplemented with 2% fetal bovine serum. Cells were incubated in an orbital shaker for 72 h at 27 °C. Cultures were centrifuged at $900 \times g$ for 10 min to harvest the supernatants containing the viruses.

### SIT1 expression and purification
Expi293F GnTI− cells in Freestyle 293 Expression Medium (Thermo Fisher Scientific) were transduced with the SIT1 P3 baculovirus (3% v/v)

in the presence of 5 mM sodium butyrate. Cells were grown in a humidity-controlled orbital shaker for 72 h at 30 °C with 8% CO$_2$ before being harvested by centrifugation at 900 g for 15 min, washed with phosphate-buffered saline, and flash-frozen in liquid nitrogen. Cell pellets were stored at −80 °C until further use.

Cell pellets expressing full-length SIT1 were resuspended in a lysis buffer of 50 mM HEPES pH 7.5, 300 mM NaCl, 1.5% glycol-diosgenin (GDN, Anatrace), and cOmplete EDTA-free Protease Inhibitor Cocktail (Roche). Lysate was loaded on TALON resin (Takara Bio) gravity flow column, washed with column buffer (50 mM HEPES pH 7.5, 300 mM NaCl, 0.02% w/v GDN) supplemented with 1 mM ATP and 10 mM MgCl$_2$, and eluted in column buffer with 300 mM imidazole. The eluent was immediately loaded on Strep-Tactin XT resin (IBA) gravity flow column, washed with column buffer supplemented with 1 mM ATP and 10 mM MgCl$_2$, and eluted in column buffer with 50 mM D-biotin. The protein was further purified by size exclusion chromatography using a Superdex 200 Increase (10/300) GL column pre-equilibrated with SEC buffer (20 mM HEPES pH 7.5, 150 mM NaCl, 0.02% w/v GDN).

### Thermostabilization measurements
Purified SIT1 was diluted to 0.4 mg/mL in SEC buffer. Amino acids at 0.5 mM were added to the protein and incubated on ice for 1 h, then Plex nanoDSF Grade High Sensitivity Capillaries (NanoTemper) were filled with 10 μl protein sample. Melting curves were determined using Prometheus NT.48 by monitoring the intrinsic fluorescence at 350 nm relative to 330 nm during a temperature ramp (1 °C/min increase) from 20 to 95 °C. The melting temperature was determined from measurements of three biological replicates.

### ACE2-SIT1 expression and purification
Expi293F GnTI− cells in Freestyle 293 Expression Medium (Thermo Fisher Scientific) were transduced with the P3 baculovirus for ACE2 and SIT1 (1.5% v/v for each virus) in the presence of 5 mM sodium butyrate. Cells were grown in a humidity-controlled orbital shaker for 72 h at 30 °C with 8% CO$_2$ before being harvested by centrifugation at 900 × g for 15 min, washed with phosphate-buffered saline, and flash-frozen in liquid nitrogen. Cell pellets were stored at −80 °C until further use.

Cell pellets expressing full-length ACE2-SIT1 complex were resuspended in a lysis buffer of 50 mM HEPES pH 7.5, 300 mM NaCl, 1.5% glycol-diosgenin (GDN, Anatrace), and cOmplete EDTA-free Protease Inhibitor Cocktail (Roche). Lysate was loaded on Strep-Tactin XT resin (IBA) gravity flow column, washed with column buffer supplemented with 1 mM ATP and 10 mM MgCl$_2$, and eluted in column buffer with 50 mM D-biotin. The eluent was immediately loaded on an anti-FLAG M2 affinity resin gravity flow column, washed with column buffer, and eluted in column buffer supplemented with 0.2 mg/mL FLAG peptide. The protein was further purified by size exclusion chromatography using a Superose 6 Increase (10/300) GL column pre-equilibrated with SEC buffer.

### Cryo-EM sample preparation and data collection
Peak fractions of purified ACE2-SIT1 complex were pooled, incubated with 10 mM L-pipecolate or 10 mM glycine, and concentrated to ~4.5 mg/mL or ~6 mg/mL. cryo-EM grids were prepared using a Mark IV Vitrobot (Thermo Fisher Scientific) by applying protein to glow-discharged QuantiFoil Au R1.2/1.3 200-mesh grids (Quantifoil), blotting for 3.0 s under 100% humidity at 4 °C, and then plunging into liquid ethane.

The pipecolate dataset was collected on a Titan Krios electron microscope, using a GIF-Quantum energy filter with a 20 eV slit width (Gatan) and a K3 direct electron detector (Gatan) at a dose rate of 19.5 e⁻/px/sec. EPU (Thermo Fisher Scientific) was used to automatically record three movie stacks per hole (super-resolution / EPU bin 2) with the defocus ranging from −1.2 to −2.4 μm. Each micrograph was dose-fractioned into 50 frames, with an accumulated dose of 50 e⁻/Å$^2$.

The glycine dataset was collected on a Titan Krios electron microscope, using a GIF-Quantum energy filter with a 5 eV slit width (Gatan) and a K2 direct electron detector (Gatan). SerialEM was used to automatically record three movie stacks per hole[69], at a dose rate of 8.42 e⁻/px/sec with the defocus ranging from −1.0 to −2.2 μm. Each micrograph was dose-fractioned into 50 frames, with an accumulated dose of 50 e⁻/Å$^2$.

### Reconstruction of ACE2-SIT1 with pipecolate
cryoSPARC was used for the majority of the data processing workflow[70], with RELION used only for final 3D classification (without alignment) for the focused refinement of the SIT1 component[71]. Movies were motion corrected and CTF-corrected in cryoSPARC.

For the pipecolate dataset, particles were blob picked, followed by two cycles of 2D classification. The well-resolved 2D classes were used for template-based picking. Ab initio models generation and heterogeneous classification yielded maps with open and closed conformation of the ACE2 peptidase domain. The particles from each conformation were separated by further classification into species with 2:2 and 2:1 stoichiometries of ACE2 and SIT1. Non-uniform refinement with C2 symmetry imposed gave reconstructions for the 2:2 ACE2:SIT1 open and closed PD conformations at 3.29 Å and 3.24 Å, respectively.

To improve the map for SIT1, all particles from the 2:2 open and closed reconstructions were aligned with C2 symmetry imposed to give a consensus 2:2 ACE2-SIT1 reconstruction. The aligned particles were symmetry-expanded and local refinement was performed within a region encompassing a single SIT1 monomer. The resultant aligned particles were then subjected to 3D classification in RELION without further alignment (K = 10, T = 12). The particles in the best 3D classes, based on estimated resolution criteria, were pooled for local refinement in cryoSPARC using the SIT1 monomer mask to produce a 3.49 Å reconstruction.

### Reconstruction of ACE2-SIT1 in glycine
For the apo dataset, particles were blob picked, followed by two cycles of 2D classification. The particles from well-resolved 2D classes were used for Topaz training and particle picking[72]. Subsequent two cycles of ab initio model generation and heterogenous refinement yielded maps with open and closed conformation of the ACE2 peptidase domain. The particles from each conformation were further classified into species with 2:2 and 2:1 stoichiometries of ACE2 and SIT1. Non-uniform refinement with C2 symmetry imposed gave reconstructions for the 2:2 ACE2:SIT1 open and closed PD conformations at 3.59 Å and 3.46 Å, respectively.

To improve the map for SIT1, all particles from the 2:2 open and closed reconstructions were aligned with C2 symmetry imposed to give a consensus 2:2 ACE2-SIT1 reconstruction. The aligned particles were symmetry-expanded and local refinement was performed within a region encompassing a single SIT1 monomer. The resultant aligned particles were then subjected to 3D classification in RELION without further alignment (K = 10, T = 12). The particles in the best 3D classes, based on estimated resolution criteria, were pooled for local refinement in cryoSPARC using the SIT1 monomer mask to produce a 3.76 Å reconstruction.

### Model building and refinement
Models were initially built for the open and closed ACE2 dimer using the pipecolate dataset. Published structures of B$^0$AT1 (PDB: 6M18), ACE2 with PD open (PDB: 6M1D), and ACE2 with PD closed (PDB: 6M18) were used as templates for model building. The B$^0$AT1-derived SIT1 model was pruned using CHAINSAW[73]. SIT1 residues 10–582 and ACE2 residues 740–768 were built using the 3.49 Å transmembrane domain-

focused map in Coot[74]. Models were refined with phenix.real_space_refine using default geometric restraints[75]. For the open and closed 2:2 ACE2-SIT1 complexes, the focused SIT1 coordinates were used as a reference model during refinement. Geometric restraints for pipecolate were generated using GRADE[76]. The pipecolate-bound ACE2 and SIT1 protein models were used as templates for subsequent model building for ACE2-SIT1 structure determined in the presence of glycine. The ACE2 components required minimal adjustments and differences in the SIT1 were primarily localized around TM1a.

## B⁰AT1-leucine refitting

The B⁰AT1-leucine complex from 6M17 was subjected to global real-space refinement against the deposited focused cryo-EM map (EMD-30043) at 3.1 Å using PHENIX. The binding mode of the substrate leucine (Leu707) was then adjusted to maximize its interactions within the binding site while remaining consistent with the cryo-EM density. Alterations were also made to B⁰AT1 substrate pocket which included flipping the peptide backbone of residues Gly51 and Leu52 as well as changes to the sidechain rotamers of Val50, Leu52, Val55 and Trp56. The refitted model was also subjected to global real-space refinement against the deposited focused cryo-EM map. Qscores were calculated for the re-refined and refitted/re-refined B⁰AT1 models using MapQ[56].

## Electrophysiology

For electrophysiology, codon-optimized genes for SIT1 and mutants were synthesized into pUC57 with a T7 promoter, 5' and 3' UTR from *Xenopus* beta globin, and a 64 nucleotide polyA tail. Plasmid DNAs were linearized with XmaI (New England Biolabs). Complementary RNA was synthesized using the mMESSAGE mMACHINE T7 transcription kit (ThermoFisher Scientific).

Stage V oocytes were isolated from the lobe via digestion with 2 mg/mL collagenase A (Boehringer) at 26 °C for 1 h and 30 ng of cRNA encoding WT or mutant SIT1 was injected into each oocyte cytoplasm (Drummond Nanoinject; Drummond Scientific Co). The oocytes were then stored for 60 h at 17 °C in frog Ringer's solution (96 mM NaCl, 2 mM KCl, 1 mM MgCl₂, 1.6 mM CaCl₂, 5 mM HEPES, pH 7.4), which was supplemented with 2.5 mM sodium pyruvate, 0.5 mM theophylline, 50 μg/mL gentamicin, and 100 μg/mL tetracycline.

Whole-cell currents generated by substrates were recorded with a Geneclamp 500 amplifier (Axon Instruments), digitized by a Powerlab 2/20 chart recorder (ADInstruments), and processed by LabChart version 8 (ADInstruments). Oocytes were clamped at −60 mV and recordings were performed in frog Ringer's solution. For selectivity assays, transport currents for L-proline, glycine, sarcosine, L-leucine, L-lysine, and L-alanine (1 mM, pH 7.4) were measured for WT and mutant SIT1 ($n = 5$). L-proline concentration (1–3000 μM) dependent transport currents were measured for WT and mutant SIT1 ($n = 5$). The Michaelis constant (Km) and maximal current generated by substrate transport (Imax) of L-proline in WT and mutant SIT1 were determined using the Michaelis–Menten model in GraphPad Prism Version 10.2.0. Data was tested for normality using the Shapiro-Wilk test in GraphPad Prism Version 10.2.0.

## Reporting summary

Further information on research design is available in the Nature Portfolio Reporting Summary linked to this article.

## Data availability

The data that support this study are available from the corresponding authors upon request. The cryo-EM maps and models generated in this study have been deposited in the EMDB database and the Protein Data Bank, respectively. The cryo-EM maps have been deposited in the Electron Microscopy Data Bank (EMDB) under accession codes EMD-17381 (ACE2-SIT1 with pipecolate bound and open peptidase domain), EMD-17382 (ACE2-SIT1 with pipecolate bound and closed peptidase domain), EMD-17380 (SIT1 with pipecolate bound focused refinement), EMD-17378 (ACE2-SIT1 with open peptidase domain determined in the presence of glycine), EMD-17379 (ACE2-SIT1 with closed peptidase domain determined in the presence of glycine), and EMD-17377 (SIT1 focused refinement determined in the presence of glycine). The atomic coordinates have been deposited in the Protein Data Bank (PDB) under accession codes 8P30 (ACE2-SIT1 with pipecolate bound and open peptidase domain), 8P31 (ACE2-SIT1 with pipecolate bound and closed peptidase domain), 8P2Z (SIT1 with pipecolate bound focused refinement), 8P2X (ACE2-SIT1 with open peptidase domain determined in the presence of glycine), 8P2Y (ACE2-SIT1 with closed peptidase domain determined in the presence of glycine), and 8P2W (SIT1 focused refinement determined in the presence of glycine). A Source Data file is available. Source data are provided with this paper.

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

## Acknowledgements

This work was financially supported by the BBSRC (BB/V018051/1) (to E.P.C. and T.D.). D.B.S., J.H., and A.C.W.P. were supported by the Innovative Medicines Initiative 2 Joint Undertaking (JU) under grant agreement No 875510. The JU receives support from the European Union's Horizon 2020 research and innovation program and EFPIA and Ontario Institute for Cancer Research, Royal Institution for the Advancement of Learning McGill University, Kungliga Tekniska Högskolan, Diamond Light Source Limited. We thank Beth Maclean for assistance with EM sample preparation and screening, Loic Carrique for helping with OPIC data collection setup, Brian Marsden for assistance with cryo-EM data processing resources, and Wyatt Yue for assistance with project management. Electron microscopy was provided through the Oxford Particle Imaging Centre (OPIC), an Instruct-ERIC centre (funded by Wellcome Trust JIF award [060208/Z/00/Z] and equipment grant [093305/Z/10/Z]) and the Electron Bio-Imaging Centre, Diamond Light Source Ltd (eBIC; BAG proposal bi28713).

## Author contributions

H.Z.L, J.S.H., S.R.B., D.W., L.S., and D.B.S. cloned SIT1 and ACE2. H.Z.L., J.S.H., S.C.L., K.E.J.R., S.R.B., D.S., A.E., D.H., and C.N. expressed and purified the proteins. H.Z.L. collected and analyzed the thermostabilization data. H.Z.L., G.C., and A.C.W.P. collected and processed the cryo-EM images and built the atomic models. H.Z.L., A.C.W.P., and D.B.S. analyzed the structures. I.L. and R.J.V. collected and analyzed the electrophysiology data. H.Z.L., A.C.W.P., and D.B.S. wrote the manuscript. All authors participated in the discussion and manuscript editing. N.A.B.B., R.J.V., T.R.D., E.P.C., and D.B.S. supervised the research.

## Competing interests

The authors declare no competing interests.
