## [Peer Review File · Nature Communications]

Structure and function of the SIT1 proline transporter in complex with the COVID-19 receptor ACE2Reviewer #1 (Remarks to the Author):

The study authored by Li et al describes the structure of the proline transporter SIT1 from the SLC6 family transporters in complex with the ACE2 that is a target of the COVID spike protein for cellular entry. The structures of SIT1 were determined with a bound substrate analogue pipecolate and in the substrate-free state in two conformations of inward-occluded state and inward-open state respectively. Insights into proline transport are vital to as L-proline serves as a signaling neurotransmitter in addition to being a vital amino acid and deficiencies in transport can cause conditions like iminoglycinuria. While the manuscript describes a biomedically important structural system the authors do not substantiate their findings through biochemical analyses that can be done using functional assays. The following are the major concerns that I could identify in this study.

1. Pipecolate density fits the imine ring but density is lacking for the carboxylate and some discussion points can be added as to why it could be the case.
2. Why are there no uptake assays with the expressed protein. The importance of Ser at Gly253, role of T199M could be better studied through some straightforward biochemical assays.
3. Can the authors discuss if there is L-Proline /D-Proline discrepancy in uptake and if yes how is this chiral selection enforced.
4. Figures can be improved by adding more details.
5. The presence of Na⁺ ions is inferred from the presence of suitable coordination geometry in the structures and their comparison to known structures.
6. The depiction of main figures 3 and 5 can improve substantially. Distances can be added to enhance information. Suppl figures 4a and/or 4h can be added to the main figure 3.
7. The authors would need to show the local resolution around the substrate binding site as the transporter region of the structures seems to have resolved to a relatively low resolution.
8. A figure needs to be added to the supplementary figures to describe the effects of the T199M congenital mutant which the authors describe in one section

Overall the study requires major revisions through the addition of biochemical data in the form of uptake assays and some mutagenesis data for the G253S and T199M mutants to enhance the conclusions of the manuscript.

Reviewer #2 (Remarks to the Author):

This is a well performed and important study. It clarifies and carefully analyzes a number of uncertainties around the substrate binding modes of SLC6 amino acid transporters and suggests a mechanism for the iminoglycinuria-associated mutation T199M.

Specific comments:

Fig. 1c; The glycans are very hard to see, perhaps using a different colour could help.

Line 153: Where does the second ACE2 bind in the 2:1 complex? This is potentially interesting because the equivalent residue to R213 is an ACE2-dependent mutation in SLC6A19 (R240Q) but does not seem to be close to any contact site. Perhaps this can be discussed.

REVIEWER COMMENTS

Reviewer #1 (Remarks to the Author):

The study authored by Li et al describes the structure of the proline transporter SIT1 from the SLC6 family transporters in complex with the ACE2 that is a target of the COVID spike protein for cellular entry. The structures of SIT1 were determined with a bound substrate analogue pipicolate and in the substrate-free state in two conformations of inward-occluded state and inward-open state respectively. Insights into proline transport are vital to as L-proline serves as a signaling neurotransmitter in addition to being a vital amino acid and deficiencies in transport can cause conditions like iminoglycinuria.

While the manuscript describes a biomedically important structural system the authors do not substantiate their findings through biochemical analyses that can be done using functional assays. The following are the major concerns that I could identify in this study.

1. Pipicolate density is fits the imine ring but density is lacking for the carboxylate and some discussion points can be added as to why it could be the case.

We thank the reviewer for their keen eye for detail and pointing out the weak density which is a well-known challenge for resolving negatively charged groups. This is likely due to the local negative charge scattering electrons more weakly, and carboxylates being more prone to radiation damage. Nevertheless, the density's shape and the local chemistry allow us to unambiguously orient pipicolate in the binding site. We have revised the manuscript to note these technical challenges.

2. Why are there no uptake assays with the expressed protein. The importance of Ser at Gly253, role of T199M could be better studied through some straightforward biochemical assays.

Reviewer #1 is quite right in pointing out that our study would be greatly enhanced by functional assays to support our hypotheses regarding the role of Gly253 and T199 in the protein's selectivity and transport cycle. Therefore, we have collaborated with experts in transporter electrophysiology, adding to the revised manuscript measurements of amino acid selectivity for SIT1 and its mutants G253S and N410A. These results greatly enrich our understanding for the critical role of these residues in SIT1's selectivity for secondary amino acids. Similarly, we have added measurements of SIT1 kinetics for wild-type protein and the mutants T199M and V196F. These show the TM1a-TM5 interaction directly regulates the transporter turnover rate (I_{max}), supporting our biophysical model for T199M effects in iminoglycinuria.

Critically, these results highlighted that secondary amino acid selectivity, and the link between substrate binding and enzyme turnover, are more nuanced than our initial hypotheses. We have revised the manuscript accordingly, and are very grateful to the reviewer for suggesting this series of experiments.

3. Can the authors discuss if there is L-Proline /D-Proline discrepancy in uptake and if yes how is this chiral selection enforced.

The reviewer raises a very intriguing question regarding the stereo-selectivity of SIT1. The transporter's selectivity for L-amino acids was previously noted by Stevens & Wright (1985). Examining our structure, D-proline and D-pipecolate would clash with several residues of the binding pocket, a mechanism that agrees with the previously proposed stereo-selectivity mechanism for LeuT-fold transporters (Ma et al. 2019). We thank the reviewer for pointing out this additional insight our structure provides, and we have revised the text to include these new insights.

4. Figures can be improved by adding more details.

The reviewer is quite correct that additional detail would improve the figures, and we have revised them accordingly. In Figure 1, we have labeled all relevant side chains in panels f and g. In Figure 2, we have added distances to the polar interacting groups of panel b, and the side chains of panels d-f. Also, we have added bond distances in Figures 3c, 3e, and 4d. We thank the reviewer for pointing out this opportunity to make the figures clearer and convey more detail.

5. The presence of Na⁺ ions is inferred from the presence of suitable coordination geometry in the structures and their comparison to known structures.

We thank the reviewer for pointing out this challenge with our dataset. This is a common problem with CryoEM maps at similar resolutions of sodium-coupled membrane transporters, and we have emphasized this point in the revised text.

6. The depiction of main figures 3 and 5 can improve substantially. Distances can be added to enhance information. Suppl figures 4a and/or 4h can be added to the main figure 3.

The reviewer raises a good point for how to improve the information density and clarity of our figures. We have revised the manuscript based on the reviewer's feedback, adding distances and moving the LeuT/MhsT/SIT1 overlay to Figure 3. While we agree that additional labels would be helpful in Figure 5, adding distances from the inferred Na1 and Na2 ions would be inappropriate as the ions' locations cannot be refined in our model. Therefore, we have revised the text to describe the displacement and movement of the ion-coordinating groups, which have been experimentally refined.

7. The authors would need to show the local resolution around the substrate binding site as the transporter region of the structures seems to have resolved to a relatively low resolution.

Reviewer #1 raises an important point, as the local resolution within the transport domain is not immediately clear from the figures. The resolution in this region is quite well resolved, with the substrate binding residues at ~3Å resolution. To clarify this, we have added Supplementary Figures 6 panels g and h, which clearly show the SIT1 binding site resolution in both pipecolate-bound and apo-states.

8. A figure needs to be added to the supplementary figures to describe the effects of the T199M congenital mutant which the authors describe in one section

We are very grateful for the reviewer's suggestion of an illustration to clarify the biophysical mechanism for the T199M mutation. We have added Supplementary Figure 7 to the revised manuscript, with an illustration of our model for the mutation's effects on the cytoplasmic gate.

Overall the study requires major revisions through the addition of biochemical data in the form of uptake assays and some mutagenesis data for the G253S and T199M mutants to enhance the conclusions of the manuscript.

We thank the reviewer for highlight these key points. We have addressed these in the revised manuscript with electrophysiology measurement of substrate selectivity and transport for V196F, T199M, G253S, and N410A, which significantly strengthens the story.

Reviewer #2 (Remarks to the Author):

This is a well performed and important study. It clarifies and carefully analyzes a number of uncertainties around the substrate binding modes of SLC6 amino acid transporters and suggests a mechanism for the iminoglycinuria-associated mutation T199M.

Specific comments:

Fig. 1c; The glycans are very hard to see, perhaps using a different colour could help.

We thank Reviewer #2 for this critical insight on how to make our figures more clear. In the revised manuscript, we have revised Figure 1c to emphasize the glycans by modifying their size and color.

Line 153: Where does the second ACE2 bind in the 2:1 complex?

The ACE2 dimer remains largely unchanged in the 2:2 and 2:1 complexes, though the ACE2 transmembrane helix cannot be resolved without being bound to SIT1. This is unsurprising as this helix is likely more flexible when not bound to the transporter. However, we do not pursue modeling this stoichiometrically asymmetric complex because we cannot identify if the loss of SIT1 is physiological, or an artifact of our experimental conditions, as discussed in the manuscript. Rather, detailed studies of SIT1-ACE2 interactions and stoichiometry are the subject of ongoing experiments in the group, but beyond the scope of this study.

This is potentially interesting because the equivalent residue to R213 is an ACE2-dependent mutation in SLC6A19 (R240Q) but does not seem to be close to any contact site. Perhaps this can be discussed.

We kindly thank the reviewer for pointing out this opportunity to examine SLC6 proteins' trafficking in complexes with ACE2. However, the mechanism for

SLC6A19's R240Q is not clear from the structure. Further, the differences in ACE2 binding to various SLC6 proteins may be distinct and will require a focused set of experiments beyond the scope of this report. Studies on SIT1's trafficking are ongoing and will be the subject of future publications. Nevertheless, in the revised manuscript we have expanded our discussion of the ACE2-SIT1 interactions to serve as a reference for future comparisons of SLC6 interactions with chaperone proteins.

Reviewer #1 (Remarks to the Author):

The revised manuscript suitably addresses all my concerns of functional assays, figures and data quality. This is now a well organised and performed study that can be published after editorial checks.

Reviewer #2 (Remarks to the Author):

The comments made by this reviewer have been addressed.